# The Anatomical Pathogenesis of Stress Urinary Incontinence in Women

**DOI:** 10.3390/medicina59010005

**Published:** 2022-12-20

**Authors:** Xunguo Yang, Xingqi Wang, Zhenhua Gao, Ling Li, Han Lin, Haifeng Wang, Hang Zhou, Daoming Tian, Quan Zhang, Jihong Shen

**Affiliations:** 1The First Department of Urology, The First Affiliated Hospital of Kunming Medical University, Kunming 650032, China; 2Yunnan Province Clinical Research Center for Chronic Kidney Disease, Kunming 650032, China

**Keywords:** SUI, anatomy, pathogenesis, female

## Abstract

Stress urinary incontinence is a common disease in middle-aged and elderly women, which seriously affects the physical and mental health of the patients. For this reason, researchers have carried out a large number of studies on stress urinary incontinence. At present, it is believed that the pathogenesis of the disease is mainly due to changes related to age, childbirth, obesity, constipation and other risk factors that induce changes in the urinary control anatomy, including the anatomical factors of the urethra itself, the anatomical factors around the urethra and the anatomical factors of the pelvic nerve. The combined actions of a variety of factors lead to the occurrence of stress urinary incontinence. This review aims to summarize the anatomical pathogenesis of stress urinary incontinence from the above three perspectives.

## 1. Introduction

Stress urinary incontinence (SUI) refers to the involuntary leakage of urine from the external urethral orifice when abdominal pressure increases due to actions such as sneezing, coughing and laughing. The prevalence of SUI among adult women in China is as high as 18.9% [1], and it seriously affects the physical and mental health of the patients and also represents a significant medical and economic burden on society [1,2]. To this end, researchers have conducted a large number of studies. Kelly observed an open vesical neck in women with SUI [3]. Enhorning proposed the theory of pressure transmission, suggesting that the intra-abdominal pressure is evenly applied to the bladder and proximal urethra, and noticed that the transmission was reduced in women with SUI [4]. Petros and Ulmsten presented an integral theory explaining why the reconstruction and restoration of urethral support can improve continence [5,6]. Delancey proposed the “hammock theory”, suggesting that the levator ani muscle, the anterior vaginal wall, the pelvic fascia and the pubourethral ligament, together, form a hammock structure to perform the function of urine control [7]. Based on these theories, a number of treatment methods for stress urinary incontinence have been proposed, and certain therapeutic effects have been achieved, but there are still many complications, including a high recurrence rate and other problems. This situation requires us to further explore the anatomic pathogenesis of stress urinary incontinence. Currently, it is believed that the pathogenesis of stress urinary incontinence is mainly due to changes related to age, childbirth, obesity, constipation and other risk factors [8,9] that induce changes in the urinary control anatomical factors, including the anatomical factors of the urethra itself, the anatomical factors of periurethra and the anatomical factors of the pelvic nerve [10]. Finally, stress incontinence occurs under the combined effects of a variety of change factors. Therefore, this review summarizes the anatomical pathogenesis of SUI from the above three perspectives so as to facilitate a better understanding of the occurrence of SUI and provide new ideas for the clinical treatment of SUI.

## 2. Anatomical Factors of the Urethra Itself

### 2.1. The Sealing Effect of the Urethral Mucosa Is Weakened

The urethral mucosa has a sealing effect. The normal urethral mucosa can produce mucus-like secretions, which create a strong sealing effect with the assistance of the submucosal blood vessels. The submucosa of the urethra is rich in blood vessels, providing a rich blood supply to the mucous membrane, promoting the proliferation of mucous membrane cells, thickening the mucous membrane, promoting mucous secretion by the mucous membrane cells and increasing the closure performance of the urethral lumen [11]. When the blood vessels in the submucosa are filled, the urethral lumen can become compressed to strengthen the sealing effect of the urethral mucosa [11]. Estrogen can act on the submucosal blood vessels of the urethra, promoting their dilation and filling and causing them to become highly vascularized in suburethra mucosa [12], thereby finally increasing the blood supply to the mucosa and strengthening the mucosal sealing effect of the urethral mucosa [13,14]. When the estrogen levels are low, the submucosal blood vessels of the urethra are reduced, the mucosal blood supply is insufficient, causing the mucosa atrophies and the urethral mucosal sealing effect to be weakened, and, finally, urinary incontinence is likely to occur [15]. In chronic urethritis, long-term inflammatory stimulation leads to urethral mucosal fibrosis, urethral mucosal atrophy, a reduction in mucus secretion and a reduction in the submucosal blood vessels, which affects the mucosal sealing effect of the urethral mucosa and easily leads to the occurrence of SUI [16] (Figure 1).

### 2.2. Dysfunction or Defect of the Urethral Sphincter

The muscular layer of the urethra is formed of urethral smooth muscle and striated muscle (external urethral sphincter) [11,17,18]. The urethral smooth muscle is divided into two layers [19]. The inner layer is a longitudinal smooth muscle fiber, which is arranged longitudinally with the urethral lumen at the center [11]. Additionally, its elastic contraction increases the diameter of the longitudinal smooth muscle bundle, effectively narrowing the urethral lumen and increasing the resistance to urination. The outer layer is a circular smooth muscle fiber that completely envelops the urethra, which is a slow-contraction muscle fiber with anti-fatigue properties [20]. The inner layer of urethra smooth muscle fibers is arranged longitudinally and circularly surrounded by the outer layer of urethra smooth muscle fibers [11]. This kind of unique tissue arrangement is very important for continence in the context of urination. The outer circular smooth muscle fiber contracts with the inner longitudinal smooth muscle as a central filler, producing continuous tension [21], narrowing the lumen diameter of the urethra and maintaining static urethral tension [11,17] (Figure 1).

The external urethral sphincter in women is Ω-shaped [22]. It is mainly distributed on the ventral surface and both sides of the urethra and does not completely envelop the urethra, forming the outermost layer of the urethra and covering about 80% of the total length of the urethra [23,24,25]. The external urethral sphincter uses the anterior vaginal wall as a “plate” to compress the urethra in the direction of the posterior urethral wall to achieve continence when the external urethral sphincter contracts [18,26]. Morgan et al. [27] studied the components of the external urethral sphincter fibers and found that female external urethral sphincter muscle fibers can be divided into slow-contractile muscle fibers and fast-contractile muscle fibers [28]. Slow-contractile muscle fibers have anti-fatigue properties, and they can continuously contract to generate tension and maintain a resting urethral pressure. When the abdominal pressure increases, the fast-contracting muscle fibers contract rapidly, compressing the urethra in the direction of the posterior wall of the urethra, preventing urine leakage and maintaining urethral pressure during stress periods [29]. Frauscher et al. [30] used ultrasound to visualize the external urethral sphincters of SUI patients and normal women. It was found that the external urethral sphincter in SUI patients was weak. Defects in, or the dysfunction of, the urethral sphincter can be divided into congenital and acquired categories. Congenital urethral sphincter disorders are mainly caused by congenital central nervous system diseases, while acquired urethral sphincter disorders are mainly caused by childbirth, surgical treatments and radiotherapy [31]. Incontinence easily occurs when the urethral sphincter cannot contract effectively [32].

### 2.3. Decreased Elasticity of the Urethral Wall

Normally, the female urethra is a soft tubular structure. The urethral wall is rich in loose connective tissue, elastic fiber, collagen and other components, so that the urethral wall has good elasticity and flexibility. Under the actions of external forces, it can effectively deform and close the urethra, ensuring the tightness of the urethral closure [33]. Zinner et al. [34] showed that, in the mechanical model, when the lumen surface is filled with lubricant, the elasticity of the pipe wall is better, and the flow resistance of the pipe cavity is greater [35]. Thus, the better the elasticity of the urethral wall is, the greater the flow resistance will be when the surface of the urethral lumen is filled with mucus. If the urethra undergoes multiple radiotherapy or surgery, the urethral wall will become stiff and the elasticity will be weakened, which will lead to a decrease in urethral closure capability, and urinary incontinence is prone to occur. The study found that the content of urethral elastic fibers and collagen in SUI patients was significantly lower than that of normal women [36]. Estrogen can selectively act on the urethral epithelium [12], promoting the growth and maturation of the urethral epithelium cells and the synthesis of collagen [15]. In the postmenopausal period, the decrease in the estrogen levels contributes to the decrease in the synthesis of elastic fibers and collagen in the urethral wall and the urethral closure function disorder, which can easily lead to the occurrence of urinary incontinence [13,14].

### 2.4. Shortened Length of the Functional Urethra

Ensuring the proper functional urethral length is the key factor in female continence. The thickest part of the urethral sphincter in the whole of the urethra is the middle part, which is the part mainly controlling urinary continence [37,38,39]. Tension-free middle urethral suspension for the treatment of SUI aims to ensure the effective functional length of the middle of the urethra when abdominal pressure increases and effectively improve the patient’s urinary continence function [38]. Pelsang et al. [40] showed that when the abdominal pressure of SUI patients increased, the bladder neck and the proximal urethra moved downward, forming a funnel shape and leading the relative length of the functional urethra to become shorter, thus causing urethral resistance, and the hydrostatic pressure of the urethra decreased, which can easily contribute to the occurrence of urinary incontinence. In the mechanical model, the longer the effective length of the lumen is, the greater the fluid resistance is [33,35]. Therefore, properly increasing the length of the functional urethra can increase the urethral resistance and improve the capacity for urinary continence.

## 3. Anatomical Factors Affecting the Urethra (Figure 2)

### 3.1. Weak Supporting Structure of the Bladder Neck

Under normal conditions, the bladder base is close to the horizontal level, and the bladder neck is closed. The bladder neck is located at the junction of the middle and lower third of the pubic symphysis, usually being higher than the pubococcygeal line (the line between the lower edge of the pubic symphysis and the coccyx tip) [40,41]. The angle between the proximal urethral axis and the horizontal tangent line of the bladder base is the posterior vesicourethral angle [42,43], which is normally between 90° and 110°. When the bladder neck is closed at a right angle, the proximal urethra has the strongest closing performance and the greatest resistance. Enhorning proposed the theory of pressure transmission, suggesting that the intra-abdominal pressure is evenly applied to the bladder neck and the proximal urethra in order to ensure that the intra-urethral pressure is always higher than the intra-bladder pressure, regardless of whether resting or increased abdominal pressure is applied, so as to maintain urine control [4]. The weakness of the supporting structure of the bladder neck can lead to the inadequate closure of the bladder neck, so that the posterior wall of the bladder neck collapses, the bladder neck moves downward, and the posterior vesicourethral angle increases or even disappears. Furthermore, the proximal urethra becomes a part of the bladder neck, and the intra-abdominal pressure cannot be evenly transmitted to the bladder and the proximal urethra, resulting in lower intraurethral pressure compared to the bladder pressure [4]. Eventually, these changes decrease the capacity for urinary continence, and urinary incontinence easily occurs when the abdominal pressure increases. In cystourethrography, it is found that the bladder neck of most SUI patients is funnel-shaped [44,45]. The bladder neck moves downward, reaching lower than the pubococcygeal line, and the posterior vesicourethral angle increases or even disappears [40,46,47]. McKinnie et al. [48] showed that the lack of estrogen leads to the reduction in the supporting strength of the lower urinary tract and the increase in the bladder neck activity, eventually leading to urinary incontinence [13,14] (Figure 3).

**Figure 2 medicina-59-00005-f002:**
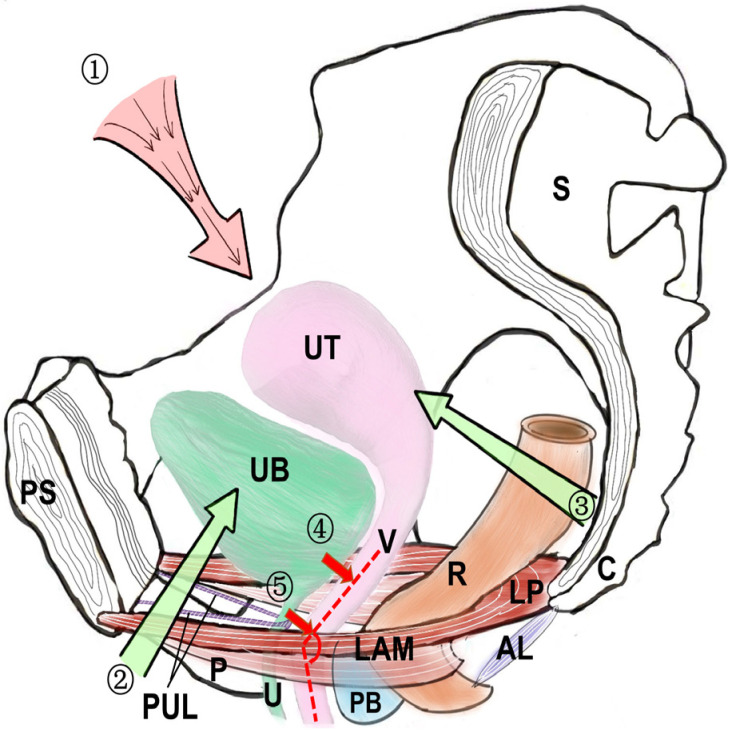
The normal pelvic floor anatomy. UB: urinary bladder, UT: uterus, V: vagina, U: urethra, R: rectum, S: sacrum, C: coccyx, PS: symphysis pubis, P: pubis PB: perineum body, LAM: levator ani muscle, PUL: pubic urethral ligament, ① upper abdominal pressure. ② Supporting force from the pubic bone. ③ supporting force from the sacrococcygeal bone. ④ Pressure from the bottom of the bladder against the upper part of the vagina ⑤ pressure from the urethra against the middle and lower parts of the vagina. Red dotted line: the angle formed by the middle and lower parts of the vagina.

### 3.2. Defective Nature or Prolapse of the Anterior Vaginal Wall Support

The middle and lower segments of the urethra cling to the anterior vaginal wall, which provides a stable “backing plate” for the middle and lower segments of the urethra. The pubococcygeal muscle contracts to support the anterior vaginal wall and squeeze the posterior urethral wall, effectively maintaining the urethral closure pressure when the abdominal pressure increases and playing a urinary control role [7]. The stability of the supporting structure of the anterior vaginal wall directly affects urine control function in women. When the anterior vaginal wall support is defective or prolapses, the posterior urethral wall support is weakened, and the patients often have different degrees of urinary incontinence when the abdominal pressure is increased [49,50].

### 3.3. The Continuity and Integrity of the Pelvic Fascia and Pelvic Fascial Tendon Arch (ATFP) Are Impaired

The intrapelvic fascia is the fibrous connective tissue that covers the surfaces of the organs in the pelvis and connects these organs with the pelvic muscles and bones [51]. It maintains the stability of the pelvic organs and allows the pelvic organs to have a certain degree of activity [52]. In particular, the intrapelvic fascia, which wraps around the side of the vagina and connects with the pelvic fascia tendinous arch, plays an important role in maintaining the stability of the vagina and urethra [53]. 

The pelvic fascial tendon arch (ATFP) is a broad-band aponeurosis structure originating from the ischial spine and ending at the ventral side of the pubis [54,55]. Its function is to suspend and fix the urethra to the anterior wall of the vagina, similar to the cable of a hammock, and maintain the stability of the urethra and vagina [7]. Pit et al. [55] found that the pelvic fascia and the pelvic fascia tendinous arch play important roles in inhibiting the backward movement of the anterior vaginal wall and the proximal urethra [54]. Delance et al. [7] believe that the anterior side wall of the vagina is connected with the pelvic fascia tendinous arch and levator ani muscle through the intrapelvic fascia. When the levator ani muscle contracts, the anterior wall of the vagina is pulled to squeeze the posterior wall of the urethra through the intrapelvic fascia, causing the urethral pressure to rise and then exerting the urinary continence function [7]. If the continuity and integrity of the pelvic fascia are destroyed, the stability of the pelvic organs such as the bladder, urethra and vagina will be weakened, and the activity of the urethra will be increased, which will easily lead to urinary incontinence.

### 3.4. Weak Pubic Urethral Ligaments

The pubourethral ligament is a dense fibrous connective tissue that connects the lower edge of the pubic symphysis with the bilateral walls of the urethra [38,56,57]. Its main function is to suspend and fix the urethra to the pubic bone, pelvic fascia tendon arch, puborectal muscle and other structures, providing a “hinge” support in the high-pressure urethral area and maintaining the stability of the urethra when the resting and abdominal pressure increases [38]. If the pubourethral ligament is weak or defective, the activity of the urethra will increase under the condition of the increase in abdominal pressure, which can easily cause urinary incontinence [56]. Kefer et al. [58] showed that in a mouse model, when the pubourethral ligament of the mouse was injured or removed, urinary incontinence occurred in the mice under the condition of the increase in abdominal pressure.

### 3.5. Levator Ani Muscle Weakness or Dysfunction

The levator ani muscle includes the iliococcygeal muscle, pubococcygeal muscle and puborectal muscle [59,60,61]. The levator ani muscle fibers are mainly composed of type I muscle fibers (slow-contraction muscle fibers) and type II muscle fibers (fast-contraction muscle fibers) [19]. The sustained contraction of the type I muscle fibers, as in the case of the “posture muscle” of the spine, pulls the middle and lower urethra, distal vagina and rectum toward the pubis [7], providing elastic support for the urethra and maintaining the normal shape and position of the urethra. Type II muscle fibers can rapidly contract when the abdominal pressure increases, causing the urethra to bend at the pelvic diaphragm, effectively closing the urethra and exerting the urinary continence function [19]. Delancey pointed out the “hammock”-like structure formed by the continuity of the intrapelvic fascia, vaginal wall and levator ani muscle. This structure can maintain urethral stability and the urethral closure pressure when the abdominal pressure increases [7]. When the levator ani muscle is weak or dysfunctional, the “hammock”-like structure cannot play an effective role, and the patient will be prone to urinary incontinence. Studies have shown that during vaginal delivery, when the fetal head passes through the urogenital hiatus, it produces great traction and shear force on the levator ani muscle [62]. This leads to the rupture and injury of the levator ani muscle, which can easily cause postpartum urinary incontinence and, later, SUI [63,64]. Stoker J et al. [65] also believe that SUI is related to levator ani muscle injury. In clinical practice, mild urinary incontinence is often treated by bioelectrical stimulation and pelvic floor muscle training so as to improve the contractility of the levator ani muscle. 

## 4. Anatomical Factors of the Pelvic Floor Nerves

### Pelvic Floor Neuromuscular Injury

The pelvic floor neuro-regulation of female continence is a complex process effected by the interaction of sympathetic, parasympathetic and somatic nerves [11,66]. The parasympathetic nerve originates from S2–S4 and mainly dominates the bladder detrusor and urethral smooth muscle. When it is excited, the nerve endings release ACH, causing the bladder detrusor to contract and the smooth muscle of the urethral and bladder neck to relax [67]. The sympathetic nerve originates from T10–L2 and also controls the bladder detrusor and smooth muscle of the bladder and urethral neck. When it is excited, the nerve endings release NE, which can act on the β receptors of the bladder detrusor and α receptors of the smooth muscle of the bladder neck and urethra, causing the relaxation of the bladder detrusor and contraction of the smooth muscle of the bladder neck and urethra [68]. The main somatic nerve involved in urination at the site of the pelvic floor is the pudendal nerve, which is issued by S2–S4 and mainly controls the external urethral sphincter. When the nerve is excited, its endings release ACH, causing the contraction of the external urethral sphincter, and when its excitability is inhibited, this causes the relaxation of the external urethral sphincter [69,70]. When nerves are damaged in any aspect, neuromuscular changes will occur, affecting normal urinary control function [71] (Figure 4).

The pelvic floor nerves are often injured by external mechanical forces. For example, when the second stage of labor is prolonged during vaginal delivery, the pelvic nerve fibers are pulled and compressed for a long time, which can easily lead to hypoxia damage. Medical procedure complications, such as the improper angle of lateral episiotomy during vaginal delivery, may cause the branches of the pudendal nerves to become damaged. After pelvic nerve injury, neuromuscular changes occur. That is, the duration of the potential nerve action is prolonged, the amplitude is reduced, the relative refractory period of potential action is prolonged, and the number of nerve fibers is significantly reduced [72]. The denervated muscle innervated by the pelvic floor nerve also gradually appears, which is characterized by muscular atrophy, a decrease in the number and density of the muscle fibers, a decrease in the diameter of the muscle fibers and a change in the proportion of muscle fiber types [73]. When the pelvic nerve is injured, the smooth muscle of the urethra, sphincter and levator ani muscle innervated by it will be denervated and atrophied, and the contractility will be weakened. When the abdominal pressure increases, the affected muscles cannot effectively contract to close the urethra and maintain the urethral closure pressure, causing urinary incontinence.

## 5. The Key Anatomical Pathogenesis and Operation Improvement

The anatomical pathogenesis of SUI is very complex, involving the anatomical factors of urethra itself, the periurethra and the pelvic floor nerve [10]. Among the many anatomical factors, the levator ani muscle and external urethral sphincter are particularly important. In a normal resting state, the contraction of the type I muscle fibers of the levator ani muscle pulls the rectum and the distal end of the vagina towards the pubic bone [7]. The levator ani muscle pulls up the anterior wall of the vagina to compress the posterior wall of the urethra under the synergistic effect of the pelvic fascia, providing a lasting elastic support to the urethra [7]. The type I muscle fibers of the external urethral sphincter contract, squeezing the urethra toward the posterior wall of the urethra and forming a force that opposes the contraction of the type I muscle fibers of the levator ani muscle [27]. The middle urethral sphincter is the thickest and has the strongest contractility. Under the joint contraction of the type I muscle fibers of the levator ani muscle and external urethral sphincter, a bend angle is formed between the middle and lower segments of the urethra, effectively maintaining the urethral closure pressure at rest and exerting the function of urinary control [19,37]. When the stress state or abdominal pressure increases, the type II muscle fibers of the levator ani muscle and the external urethral sphincter rapidly contract (the direction of the contraction is the same as that at rest) to provide strong muscle tension, leading to the upward movement of the pelvic diaphragm, the reduction in the urethral bend angle and the increase in the urethral closure pressure. Eventually, the urethra is forcefully closed to prevent urine leakage [37]. When the levator ani muscle and external sphincter of the urethra are dysfunctional, the urethra cannot be closed forcefully, resulting in urinary incontinence [74]. Therefore, in the treatment of urinary incontinence, it is very important to pay attention to the repair of the levator ani muscle and external urethral sphincter. For mild SUI, pelvic floor muscle training and bioelectrical stimulation can be used to repair the function of the levator ani muscle and external urethral sphincter. For moderate or severe stress urinary incontinence, the authors’ research team performed posterior pelvic floor reconstruction while performing anti-incontinence surgery. On both sides of Deno’s space, fishbone sutures were used to suture the levator muscle plate, bilateral vaginal wall and damaged levator ani muscle in a layer-by-layer manner from deep to shallow. This not only restores the contractility of the levator ani muscle but also effectively reduces the hiatus of the levator ani muscle, achieving a good therapeutic effect.

The supporting structure of the bladder neck in patients with SUI is weak. During cystourethrography for patients with SUI, we found that the posterior wall of the bladder neck was collapsed, while there was inadequate closure of the bladder neck, and the bladder neck had moved downward [40,46]. Furthermore, in SUI, the proximal urethra becomes a part of the bladder neck, showing a funnel shape [44,45]. The authors’ research team attempted to design an incontinence sling with an “inverted T” shape (Figure 5), placing it over the area from the posterior of the bladder neck to the posterior of the middle urethra with as little tension as possible. When the abdominal pressure increased, it effectively strengthened the bladder neck and middle urethra support and increased the length of the functional urethra. This new sling achieved good results in the treatment of SUI.

## 6. Conclusions

To sum up, the anatomical factors of the urethra itself, the periurethra and pelvic floor nerves are all very important in the anatomical pathogenesis of SUI. However, among the various anatomical factors, we believe that the joint contraction of the levator ani muscle and the external urethral sphincter, leading to the formation of the urethral bend angle and to the urethra being forcefully closed, plays a key role in urinary continence. Therefore, in the treatment of SUI, we should pay attention to the repair and reconstruction of the levator ani muscle and external urethral sphincter on the basis of the current theory of mid-urethral suspension. Secondly, the surgical approach should be improved according to the anatomical pathogenesis of SUI, and more appropriate slings, such as the “inverted T”-shaped sling (as Figure 5), should be designed to significantly improve the therapeutic effect of SUI treatments.

## Figures and Tables

**Figure 1 medicina-59-00005-f001:**
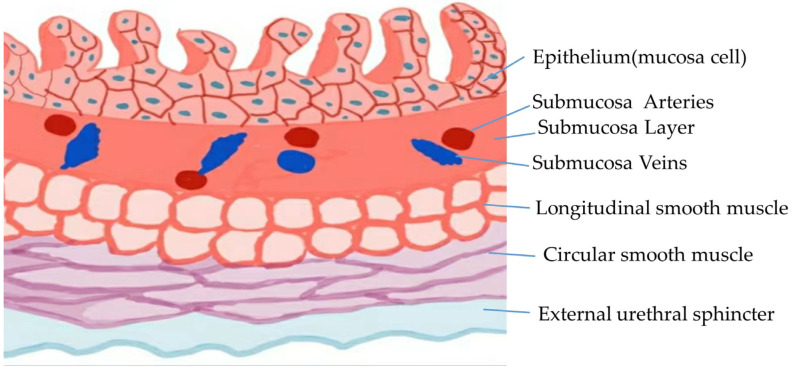
The normal transverse section of the urethral anatomy.

**Figure 3 medicina-59-00005-f003:**
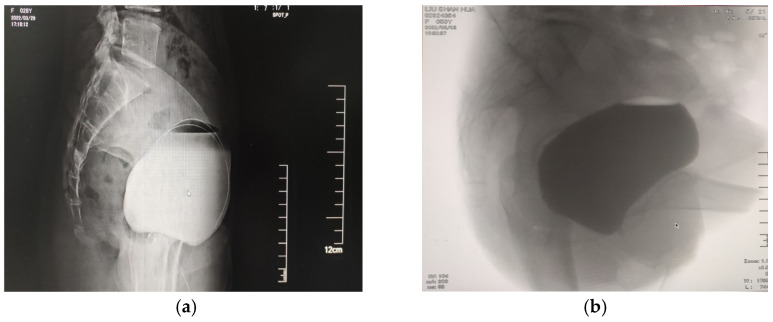
The female cystourethrography. (**a**) Normal female bladder: the bladder base is close to the horizontal level, and the bladder neck is closed. The bladder neck is located at the junction of the middle and lower 1/3 of the pubic symphysis, usually higher than the pubococcygeal line (the line between the lower edge of the pubic symphysis and the coccyx tip); (**b**) SUI patient: in cystourethrography, it is found that the bladder neck of most SUI patients is funnel-shaped. The bladder neck moves downward, reaching lower than the pubococcygeal line, and the posterior vesicourethral angle increases or even disappears.

**Figure 4 medicina-59-00005-f004:**
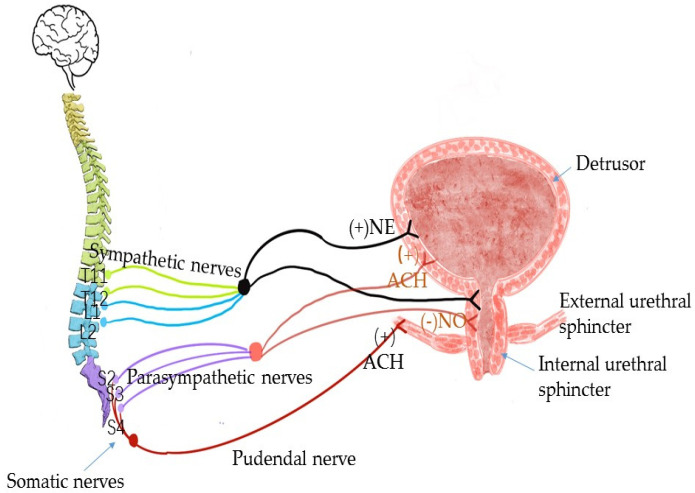
The main pelvic floor neuro-regulation of female continence.

**Figure 5 medicina-59-00005-f005:**
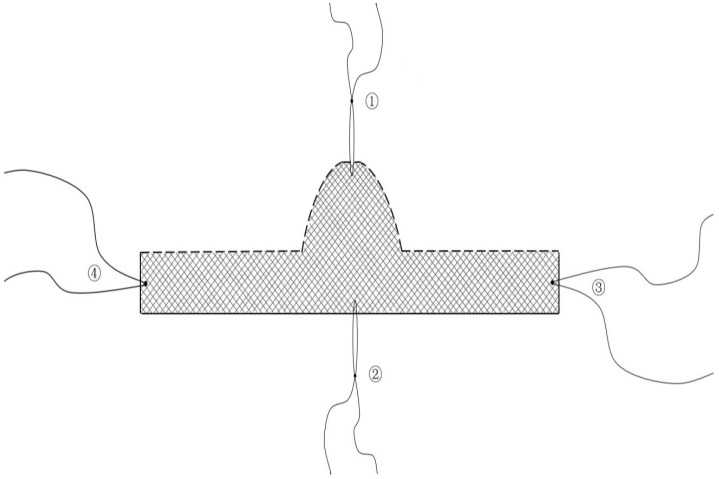
The “inverted T” shape sling: ① and ② are the central positioning lines, and ③ and ④ are the suspension guidelines on both sides.

## Data Availability

Not applicable.

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
