# Peer review of "The Anatomical Pathogenesis of Stress Urinary Incontinence in Women"

_medicina, 2022, doi:10.3390/medicina59010005_

Round 1

Reviewer 1 Report (Previous Reviewer 1)

The authors had addressed most of my previous comments. However, some points should be addressed as follows:

1. Page 9: The heading "Discussion" should be replaced by another heading because usually there is no Discussion in the Review Articles

2. The sentence "This is a picture about" should be removed from all figure legends

Author Response

Reviewer 2 Report (Previous Reviewer 2)

The manuscript is improved considerably.

Author Response

Dear reviewer,

It is my great honor to get your review and revision comments, and I would like to express my sincerely thanks to you here.

Best wishes

This manuscript is a resubmission of an earlier submission. The following is a list of the peer review reports and author responses from that submission.

Round 1

Reviewer 1 Report

This review article discussed the anatomical pathogenesis of stress urinary incontinence in women. There are some concerns in this manuscript as follows:

1.    The novel points in this review article should be clarified because there are previous reviews that discussed similar issues; e.g. https://www.ncbi.nlm.nih.gov/pmc/articles/PMC8053188/, https://www.ncbi.nlm.nih.gov/pmc/articles/PMC9306741/

2.    The “Introduction” has only 2 references. Please, add more relevant references.

3.    The “Introduction” is too short and doesn’t give the essential background information needed regarding the review subject. Please, revise.

4.    Page 2 Lines 62-66: The sentence “The muscular layer of the urethra is made up of smooth urethral muscle and striated muscle (external urethral sphincter). Urethral smooth muscle is divided into two layers. The inner layer is a longitudinal smooth muscle fiber, which is arranged longitudinally with the urethral lumen as the center. And its elastic contraction increases the diameter of the longitudinal smooth muscle bundle, effectively narrowing the urethral lumen and increasing the resistance to urination.” Needs references. Please, revise

5.    Page 2: The legend of figure 1 should be revised and modified.

6.    Page 3 Line 117: The sentence “Pelsang, R.E et al[21].” should be replaced with “Pelsang et al. [21]”

7.    Page 3 Lines 121-128: The sentence “In the mechanical model, the longer the effective length of the lumen is, the greater the fluid resistance is. Therefore, properly increasing the length of the functional urethra can increase the urethral resistance and improve the ability to urinary continence. When treating moderate or severe stress urinary incontinence, our research team adopts the “凸” shaped patch design method to make the patch cover from the bladder neck and proximal urethra to the middle urethra without tension as much as possible, effectively increasing the length of functional urethra, and achieving good treatment results.” has no references. Please, add references.

8.    Page 4: Most of the text written has only one reference; i.e. Ref. 21. Please, add more references.

9.    Page 5 Lines 176-181 had no references. Please, add.

10. I think that the conclusion should be summarized to focus on the possible clinical implications of the data obtained from the present review.

11. Page 7 Lines 287-290: The sentence “Repair and reconstruction of levator ani muscle of posterior pelvic floor: Repair the damaged levator ani muscle and perineum body by suturing, which can tighten the levator ani muscle fissure, restore the functions of levator ani muscle and perineum body, thus improving urinary control and treating stress urinary incontinence.” Should be reorganized.

12. The manuscript should be thoroughly checked regarding the grammatical and typing errors.

Reviewer 2 Report

This review aims to summarize the anatomical pathogenesis of stress urinary incontinence from the above three aspects of the anatomical factors of the urethra itself, the anatomical factors around the urethra, and the anatomical factors of the pelvic nerve

The authors comprehensively discuss the anatomical pathogenesis of stress urinary incontinence, however, there are many theories in the pathophysiology of SUI such as pressure transmission, sphincteric dysfunction, and Classification of Sphincter Weakness such as congenital or acquired should be discussed. In addition, the authors should describe what did this study add to our knowledge? 

Round 2

Reviewer 1 Report

The authors had appropriately addressed my comments.

Reviewer 2 Report

None